# Insights on the Quest for the Structure–Function Relationship of the Mitochondrial Pyruvate Carrier

**DOI:** 10.3390/biology9110407

**Published:** 2020-11-19

**Authors:** José Edwin Neciosup Quesñay, Naomi L. Pollock, Raghavendra Sashi Krishna Nagampalli, Sarah C. Lee, Vijayakumar Balakrishnan, Sandra Martha Gomes Dias, Isabel Moraes, Tim R. Dafforn, Andre Luis Berteli Ambrosio

**Affiliations:** 1Brazilian Biosciences National Laboratory, Center for Research in Energy and Materials, Campinas 13083-970, Brazil; edwin19_7@hotmail.com (J.E.N.Q.); nagampalli.krishna@monash.edu (R.S.K.N.); sandra.dias@lnbio.cnpem.br (S.M.G.D.); 2Postgraduate program in Biosciences and Technology of Bioactive Products, Institute of Biology, University of Campinas, Campinas 13083-970, Brazil; 3School of Biosciences, University of Birmingham, Birmingham B15 2TT, UK; N.Pollock@bham.ac.uk (N.L.P.); S.Lee.5@bham.ac.uk (S.C.L.); t.r.dafforn@bham.ac.uk (T.R.D.); 4São Carlos Institute of Physics, University of São Paulo, São Carlos 13563-120, Brazil; vbalakrishn5@wisc.edu; 5Membrane Protein Laboratory, Diamond Light Source Limited, Harwell Science and Innovation Campus, Didcot OX11 0DE, UK; isabel.moraes@npl.co.uk

**Keywords:** mitochondrial pyruvate carrier, membrane proteins, structure, function

## Abstract

**Simple Summary:**

The atomic structure of a biological macromolecule determines its function. Discovering how one or more amino acid chains fold and interact to form a protein complex is critical, from understanding the most fundamental cellular processes to developing new therapies. However, this is far from a straightforward task, especially when studying a membrane protein. The functional link between the oligomeric state and complex composition of the proteins involved in the active mitochondrial transport of cytosolic pyruvate is a decades-old question but remains urgent. We present a brief historical review beginning with the identification of the so-called mitochondrial pyruvate carrier (MPC) proteins, followed by a rigorous conceptual analysis of technical approaches in more recent biochemical studies that seek to isolate and reconstitute the functional MPC complex(es) *in vitro*. We correlate these studies with early kinetic observations and current experimental and computational knowledge to assess their main contributions, identify gaps, resolve ambiguities, and better define the research goal.

**Abstract:**

The molecular identity of the mitochondrial pyruvate carrier (MPC) was presented in 2012, forty years after the active transport of cytosolic pyruvate into the mitochondrial matrix was first demonstrated. An impressive amount of *in vivo* and *in vitro* studies has since revealed an unexpected interplay between one, two, or even three protein subunits defining different functional MPC assemblies in a metabolic-specific context. These have clear implications in cell homeostasis and disease, and on the development of future therapies. Despite intensive efforts by different research groups using state-of-the-art computational tools and experimental techniques, MPCs’ structure-based mechanism remains elusive. Here, we review the current state of knowledge concerning MPCs’ molecular structures by examining both earlier and recent studies and presenting novel data to identify the regulatory, structural, and core transport activities to each of the known MPC subunits. We also discuss the potential application of cryogenic electron microscopy (cryo-EM) studies of MPC reconstituted into nanodiscs of synthetic copolymers for solving human MPC2.

## 1. Introduction

The importance and versatility of pyruvate metabolism in healthy cells and disease have been the subject of extensive study for almost a century [1,2,3,4]. Anabolic and catabolic pathways that link pyruvate to energy generation and biosynthesis are diverse, depend on varying external conditions, and may frequently require physically and chemically distinct environments; the latter is achieved by the eukaryotic cell via compartmentalization [5,6]. Preceding the citric acid cycle and respiration, the oxidative decarboxylation of cytosolic pyruvate in the mitochondrial matrix requires an active proton-coupled, protein-mediated transport of this substrate from the perimitochondrial space through a heavily folded phospholipid bilayer termed the inner mitochondrial membrane (IMM) [7,8,9,10].

Since this was first reported in the early 1970s, extensive studies on putative pyruvate transporters’ physiological and mechanistic properties in mammals and plants have been undertaken. This has resulted in many functional predictions on the protein structure and mechanisms of substrate binding and transport, as well as inhibition by small molecules, as compiled by Dugan and colleagues [11]. Of particular note is the initial proposition by Nałȩcz in 1994 of a step-by-step alternating access mechanism of pyruvate transport [12]. However, the molecular identities of the proteins involved in the selective transport of pyruvate across the IMM remained elusive until 2012, when the protein sequences were concurrently presented for yeast, fruit fly, and humans; these proteins were collectively named the mitochondrial pyruvate carrier (MPC) [13,14]. 

Two systematic reviews of the vast physiology and clinical importance of MPC were recently presented [15,16]. Albeit briefly, the current knowledge on the structure of MPC—which remains mostly hypothetical owing to underlying technical difficulties—was also examined. Now, by outlining a historical perspective and a contextual discussion, we seek to further the pursuit of detailed data on the molecular mechanism of pyruvate transport by MPC. This involves computational predictions, evidence from cellular biology, biochemistry, biophysics, and efforts to isolate and reconstitute the functional complex(es) *in vitro*. Ultimately, fully realizing this goal will require high-resolution three-dimensional structures of MPC.

## 2. A Historical Perspective

Humans have two MPC subunits: MPC1 and MPC2. Both are small integral components of the IMM, of 12 and 14 kDa, respectively. Notably, an MPC1-like variant was found uniquely in placental mammals, sharing high sequence, structural, and topological identities with MPC1 [17], and a hypomorphic truncated MPC2 variant was identified in mice [18]. For simplicity, only the canonical MPC1 (UniProt Q9Y5U8) and MPC2 (UniProt O95563) will be discussed in this review. Secondary structure prediction and hydrophobicity profiling at first suggested a similar triple transmembrane helix topology (3TMH) for MPC1 and MPC2 [14]. Moreover, co-immunoprecipitation and native state electrophoresis led to the hypothesis that MPC1 and MPC2 would function together via the formation of an oligomeric complex that migrated at approximately 150 kDa [13] compared to a commercial size marker. In 2012, a distant evolutionary relationship between MPC and the bacterial sugar efflux transporter SemiSWEET was proposed by Jézégou and colleagues to reflect both an analogous 3TMH topology and a solute carrier function [19,20]. Like MPC1 and MPC2, SemiSWEET is small (around 100 residues in length).

Several atomic structures of the bacterial SemiSWEET homodimer were solved by X-ray crystallography in 2014 [21]. Representing different conformational states, these have demonstrated an alternating access mechanism of sugar transport [22]. To reconcile this finding with the proposed homology between SemiSWEET and MPC, Vanderperre and collaborators have suggested that the earlier model of large MPC oligomers [13] could be an artifact of a protein–lipid–detergent complex. Instead, they proposed a pseudo-symmetrical six-helix dimer arrangement for MPC1 and MPC2 [23]. The necessity of an asymmetrical heterodimeric MPC was justified by the asymmetry of the pyruvate molecule (to be co-transported with a proton). This contrasts with the fully symmetrical monosaccharide substrate, such as glucose, transported by the SemiSWEET homodimer [20]. The authors also noted that despite the low sequence identity between MPCs and SemiSWEET (less than 10%), a few aromatic gating residues were possibly conserved. These were not located in the central substrate-binding site, which would likely be asymmetric in MPC [23]. Of note, a bacterial SemiSWEET could promote the transport of sucrose (an asymmetric disaccharide) in an artificial *in vivo* system though the physiological relevance of this remains to be determined [24].

In 2015, a cysteine-labeling experiment by site-directed mutagenesis was used to prove that yeast MPC1 has only two membrane-spanning helices (2TMH) in isolated mitochondria [25], as opposed to the original *in silico* prediction of 3TMH [14]. Yeast contains two additional mitochondrial pyruvate carriers: MPC2 and MPC3. Both can associate with the MPC1 counterpart in a metabolic- specific context, and the assembly is dependent on the active pathway for ATP production from glucose. MPC1 and MPC2 associate during fermentation, while MPC1 and MPC3 associate during respiration [13,14,25]. Yeast MPC3 and MPC2 share, respectively, 43 and 40% sequence identity with human MPC2. Importantly, experimental data confirmed that MPC3 has 3TMH [25]. Given its relatively high sequence identity with human MPC2, this could point towards a similar 3TMH arrangement in the human protein. The orientation of the yeast protomers in the IMM were also established by Bender and colleagues: the amino and carboxy termini of yeast MPC1, and the N-terminal of MPC3 are associated with the mitochondrial matrix, while its C-terminal is associated with the intermembrane space. These results suggested a parallel arrangement between the yeast MPC1 and MPC3 molecules, like that observed for bacterial SemiSWEET [21]. 

In 2017, Dugan and colleagues presented a series of homology modeling studies [11], which attempted to reconcile the empirical knowledge on MPC accumulated before and after the pivotal works of 2012 [13,14], based on the proposed evolutionary relationship and available crystal structures of SemiSWEET [12,19,21]. These structural models considered human and plant MPC1:MPC2 heterodimers, showing both an alternating access mechanism and the structural basis of the earlier cysteine-modifying inhibitors. Other novel and improved classes of small molecule blockers were also considered [11,26,27]. As chronicled by Schell and Rutter [28] small-molecule inhibitors have been a fundamental tool for investigation in the history of MPC, from the basic research standpoint in the early 1970s to the current development of MPC-oriented therapies. 

To complement the rapidly growing body of data on MPCs’ cellular and clinical aspects, the isolation of MPCs reconstituted into artificial vesicles was considered the definitive method by which to measure MPC-dependent pyruvate transport [23,29,30]. This requires purification to the homogeneity of the functional assembly of MPC. Such a framework would allow for unprecedented biochemical, biophysical, and structural characterization, such as by X-ray crystallography or cryogenic electron microscopy (cryo-EM).

In early 2018, our group provided the first results of that endeavor [31]. We undertook heterologous co-expression and detergent-based purification strategies that were expected to reiterate previous findings and isolate the MPC1:MPC2 150 kDa complex [13]. Surprisingly, this strategy resulted in the purification of the individual MPC components in the presence of detergent at apparent molecular weights of 65 to 70 kDa. Remarkably, MPC1 was successfully purified but co-eluted with a host chaperone and mere traces of MPC2. No *in vitro* transport activity was observed for MPC1 alone. On the other hand, functional activity was detected for the MPC2 purified to homogeneity in the total absence of the MPC1 or a chaperone; again, MPC1 was co-expressed with MPC2.

We went on to provide the novel finding that human MPC2 could function as an autonomous pyruvate transporter *in vitro*, possibly in a high-order oligomeric form when in a lipid bilayer. We measured a time-dependent and substrate-specific process that relied on a proton gradient and redox balance and was sensitive to chemical inhibition by known MPC inhibitors. Notably, these properties concur with previous findings in cells or isolated mitochondria [7,10,32,33,34]. Both in a yeast cell model and in yeast isolated mitochondria that lacked endogenous MPCs we showed that human MPC2 could function as an independent transporter while stimulating cell growth and enhancing energy metabolism [31].

Employing similar experimental approaches to express and purify recombinant yeast MPC, in 2019, Tavoulari and co-workers successfully reported the isolation of both the MPC1:MPC3 heterodimer and the MPC3 homodimer in a lipid–detergent particle. The oligomeric state was determined by size-exclusion chromatography coupled to a multiangle light scattering (SEC–MALS) system [35]. Yeast MPC1 was also purified in a lipid–detergent mixture but was not stable for stoichiometric studies. Notably, the MPC1:MPC3 heterodimer, but not the MPC3 homodimer, was shown to promote rapid pyruvate transport in an artificial vesicle system with the expected functional properties [7,10,32,33,34].

Earlier in 2020, Lee and colleagues separately over-expressed human MPC1 and MPC2 in insect cells to purify not only heterodimers but also homodimers of MPC [36]. Based on SDS-PAGE, they claimed to have isolated both configurations of MPC in a complex with detergent, which allowed for stability studies by circular dichroism analysis. However, the proposed dimer stoichiometries were based solely on UV absorbance. The authors also presented computational models for MPC homo- and heterodimers based on SemiSWEET to conclude that MPC1:MPC2 is more stable and has an improved affinity to the substrate and inhibitors when compared to MPC2:MPC2. Transport studies for the isolated complexes in a lipidic vesicle system were not reported [36]. 

Additionally, in 2020, Medrano-Soto and colleagues proposed the categorization of MPC into the Transporter-Opsin-G Protein-coupled Receptor (TOG) superfamily along with Organo-Arsenical Exporter (ArsP), Endoplasmic Reticulum Retention Receptor (KDELR), L-Alanine Exporter (AlaE), and the Lipid-linked Sugar Translocase (LST). Of these new TOG superfamily members, only MPCs catalyzed substrate uptake (cation symport), while the others catalyzed the export (cation antiport). The authors identified putative transmembrane helices of MPC arranged in a 3 + 1 + 3 topology sharing similarity to the transmembrane helices 1 to 3 and 5 to 7 of eukaryotic SWEET proteins. Moreover, a hypothesis was formulated in which the MPC family originated from a 4TMH ancestor with the loss of the N-terminal TMH [37].

Finally, the MPCs’ integration into the broad canonical family of the mitochondrial carriers (MCF), also known as the solute carrier family 25 (SLC25), by primary sequence analysis alone is not possible due to the dissimilarity between the sequences. Nevertheless, since the criteria for inclusion of a family into the SLC group is not evolutionary but functional [38], MPC defines a family of its own: SLC54A [14]. Therefore, the detailed description of MPC’s still vague structure–function relationship is also of undisputed importance in the general framework of transport biology.

In summary, Figure 1 illustrates the projected structural and topological features of human MPC1 and MPC2, based on accumulated experimental knowledge, correlational analysis, and *in silico* predictions, all of which will be discussed in the next session.

## 3. Contextual Discussion

The production of pure and homogenous integral membrane proteins for functional and structural studies relies on consecutive technical approaches and choices which will be specifically discussed forward. The most common variables are the type of host cells (bacteria, yeast, insect, or mammalian cells) and the use of fusion tags to facilitate purification. Tags can differ in size, purpose, and position (N- or C-terminus). Considerations when selecting purification tags include the localization and chemical composition of the native membrane, the requirement of the translocation machinery, and the presence of a signal peptide, among others [47,48].

Three publications, reviewed in the preceding section, have reported the expression and purification of MPC. In the case of our group [31] and Tavoulari et al. [35], the human and yeast MPC, respectively, were produced by making use of the same yeast strain, which was modified (triple deleted) to lack endogenous MPC (Δmpc1/2/3) [13,14]. Both groups also used the same pBEVY-GU plasmid to allow galactose-inducible co-expression of the two (purportedly) partnering MPC subunits. In both cases, the MPC subunits were fused with a cleavable C-terminal poly-histidine tag (His-tag). For the human MPC2, an additional EGFP (enhanced green fluorescent protein) tag was added before the His-tag. The use of the C-terminal EGFP tag was widespread for membrane proteins [49,50], including the purification of large multiprotein complexes [51,52], and was particularly useful in detecting inducible expression levels, solubility, stability, and the cellular compartmentalization of recombinant MPC [31]. 

By contrast, Lee and co-workers employed conventional baculovirus plasmids co-infected in insect cells to express the individual human MPC proteins. In this case, MPC1 was fused at the C-terminal to a cleavable His-tag followed by a FLAG sequence, while MPC2 was tagged at its N-terminal either to a cleavable standard His-tag or to a longer poly-His + T4 lysozyme tag [36]. In this study, MPC1 tagged at its N-terminal and showed low expression. The common features of these three successful independent strategies are the use of eukaryotic host cells and the choice for the poly-His tag, sometimes at opposing ends. To our knowledge, there are no reports on the production of recombinant MPC using bacterial strains.

Another critical step during membrane protein production is its extraction from the native biological membrane. Thus, after inducible expression, cell sedimentation and lysis, buffered detergents (non-ionic or Zwitterionic) at concentrations several-fold above the detergent critical micelle concentration (about 1% *w/v*) are used to solubilize the membrane. At such high concentrations, detergent micelles can disrupt the lipid bilayer, surround the protein of interest, and form a soluble entity known as protein–detergent complex (PDC). The selection of the best detergent for extracting a particular membrane protein requires an empirical approach. Furthermore, the best detergent for protein solubilization is not always the most suitable for the purification steps. Often, a detergent exchange is necessary. 

The search for a buffer:detergent mixture that facilitates optimal protein extraction and purification without loss of stability, structural integrity, and function is often an expensive and lengthy trial-and-error process, especially for eukaryotic membrane proteins [53]. Hence, many high-throughput screening strategies have been proposed to attempt to streamline the process [54]. Notwithstanding this, the chances of success can be substantially improved when detergents are chosen based on previous successful experiences accumulated in ever-growing databases (https://blanco.biomol.uci.edu/mpstruc/).

In our work, although recombinant MPC1 was expressed at similar levels to MPC2, and both human proteins were co-localized in the mitochondria of the yeast host cell, we only could extract the MPC2 protein alone using octaethylene glycol monododecyl ether (C12E8). This was exchanged for n-dodecyl-β-D-maltoside (DDM) later in the course of purification [31]. Unpublished SEC–MALS studies showed that MPC2 was a monomer associated with DDM (PDC size of 57 kDa, with a protein contribution of 24%). It should be noted that MPC2 was extracted from the membrane pellet as a fusion with the EGFP-10xHis tag, which was removed by controlled proteolysis before size exclusion purification [31]. Despite a sufficiently long linker (that includes the protease cleavage site) between the hydrophilic and supposedly disordered C-terminal region MPC2, and the fused tag, we cannot exclude the possibility that the relatively larger size of this tag could disrupt the heterocomplex interface at the earlier stages of detergent-based extraction from the membranes, thus not allowing the co-purification of MPC1 and MPC2. Therefore, further investigation of the effects of the C-terminal EGFP tag on MPC complex formation is necessary. 

Lee and colleagues recently reported the extraction and purification of human MPC1 and MPC2, likewise in the detergent DDM [36]. Based on previous successes, including many high-resolution structures, DDM is historically considered the universal choice of detergent for α-helical membrane proteins [55]. The individual proteins were later mixed in different combinations to a mass ratio of 1:1 and appeared to form stable hetero- and homodimers in the micelles. The authors inferred the stoichiometry of the MPC complexes utilizing differential migration in an uncalibrated size-exclusion column, tracked only by UV absorbance [36]. It is well accepted that in the case of membrane proteins, the detergent content in a protein–detergent mixture can only be determined if the specific refractive index increment for the sample–solvent combination is tracked in parallel to both UV absorbance and right- or multiangle light scattering, which is done in SEC–MALS [56]. Hence, the stoichiometry of MPC reported in this study should be treated cautiously.

Tavoulari and co-workers achieved the purification of the recombinant yeast MPC1 and MPC3 using lauryl maltose neopentyl glycol (LMNG) from the novel class of glycol detergents, supplemented with cardiolipin [35], a signature lipid component of the IMM. Remarkably, the addition of cardiolipin to protein–detergent complexes is regarded as key for the activity and structural stabilization of numerous oligomeric and multimeric mitochondrial proteins [57]. SEC–MALS was used to infer with high confidence that either heterodimers or homodimers were present within the MPC–LMNG–cardiolipin particles. Despite the successful isolation of dimers in LMNG–cardiolipin mixture, there is no mention of the experimental outcomes in the absence of cardiolipin and how they would compare to the published data [35]. Neither we nor, as far as we can judge, Lee et al. tested cardiolipin in the purification of human MPC [31,36]; thus, the effects of novel and optimized detergents, as well as cardiolipin, on human MPC remain open to research.

The production of homogeneous recombinant membrane proteins, even in detergent environments or other native membrane mimetic systems, are no doubt milestones that enable structural and mechanistic studies [58,59]. However, fully functional characterization, especially of solute carriers, requires the reconstitution of the purified proteins back into lipid bilayer systems, often in the form of small unilamellar vesicles (SUV). This enables substrate internalization to be objectively quantified [60]. Furthermore, the formation of stable and homogeneous diffusion-free proteolipossomes is critical, where their lipid composition directly impacts vesicle stability and protein activity [60,61].

We demonstrated that the human MPC2 homomer promoted a parameter-dependent pyruvate transport when reconstituted into soy asolectin vesicles and was subjected to an electrochemical gradient [31]. As noted earlier [31], the choice for asolectin was based on the successful extraction, partial purification, and the functional reconstitution of a putative mitochondrial pyruvate carrier by Nałȩcz and co-workers, in 1986 [62]. According to the manufacturer specifications (Sigma-Aldrich #11145, St. Louis, MO, USA), asolectin is a mixture (in decreasing order) of phosphatidylcholine, phosphatidylethanolamine, phosphatidic acid, and L-α-lysophosphatidylcholine. It does not contain cardiolipin, but its precursor phosphatidic acid is present in the mixture. We also tested MPC2 reconstitution into liposomes made either from *E. coli* lipid extract or a mixture of dipalmitoylphosphatidylcholine (DPPC) and cardiolipin (unpublished data). Both failed to produce stable and monodisperse SUV at reproducible concentrations when checked by nanoparticle tracking analysis (unpublished data). In contrast, reconstitution into asolectin was successful (supporting Figure 2A in [31]). We did not perform a comparative transport assessment between the hetero-complex MPC1:MPC2 and MPC2 homomers in the *in vivo* yeast system and yeast isolated mitochondria.

The yeast MPC1:MPC3 heterocomplex, purified in LMNG:cardiolipin, also promoted substantial pyruvate transport *in vitro*, in a vesicle system composed by egg L-α-phosphatidylcholine and cardiolipin in a 20:1 (*w/w*) ratio. On the other hand, an MPC3 homomer alone failed to work under the same lipid conditions [35]. The authors reported no studies on the impact of different lipid compositions for the stability of proteoliposomes or the activity of both complexes.

The functional properties of the liposome reconstitution of human and yeast MPC [31,35] can be compared to a benchmark transport study by Halestrap, which employed isolated mitochondria from rat liver [7]. Although a pH gradient was necessary to measure transport in all studies, equilibrium was reached in as early as ten seconds for yeast MPC1:MPC3, during a time course homo-exchange assay [35]. On the other hand, around two to three minutes were necessary to achieve half-saturation into initially empty vesicles reconstituted with human MPC2 [31], which is the same time window measured for the rat liver mitochondria system [7]. The experimental K_m_ for pyruvate was higher for human MPC2, at 1.1 mM, compared to the yeast heterocomplex and the isolated rat liver mitochondria, which were more similar, at 0.3 and 0.15 mM, respectively. The maximum transport rates can also be compared. V_max_ for the yeast MPC1:MPC3 was in the range of 8 µmol.min^−1^ per milligram of purified protein, in contrast to 0.5 nmol.min^−1^ per milligram of total mitochondrial protein, in the case of rat liver mitochondrial extracts. The maximum velocity for human MPC2 was 12 nmol.min^−1^ per milligram of purified protein (calculated in retrospect) [31]. Therefore, while human MPC2 and yeast MPC1:MPC3 displayed kinetic parameters similar to those observed in mammalian mitochondria, the yeast heterocomplex was more efficient than the human homomer reconstituted vesicles, although the lipid composition of the vesicles was substantially different.

The MPC-dependent pyruvate transport is expected to be inhibited by monocarboxylate mimetics cinnamates, such as UK5099, in a mechanism known to be reliant on a covalent link to a cysteine thiol group on the transporter, reversible by the addition of reducing agents [7,8]. Accordingly, Tavoulari and colleagues reported the inhibition of yeast heterocomplex by UK5099 [35], although at inhibitor concentrations 100-fold higher when compared to studies in rat liver mitochondria [7]; the authors plausibly argue that this may reflect differential inhibitor binding between yeast and mammalian complexes [35].

On the other hand, we reported the total absence of the inhibitory activity of UK5099 over human MPC2, possibly due to the continuous presence of a reducing agent across protein preparation [31]. Notably, we showed that the preservation of the redox status of the residue Cys54 positively correlated with the functional integrity of MPC2. Noteworthy, MPC1 was originally proposed to be the bona fide target of cinnamates in yeast [13]. However, more recently, Yamashita and colleagues showed that Cys54 in human MPC2 interacts with UK5099 and α-chloroacetamide derivatives in a covalent reversible manner [63]. Most importantly, the authors did not detect the binding of UK5099 to human MPC1.

In addition, the reconstituted yeast and human MPC [31,35] complexes were also inhibited by rosiglitazone, which is a thiazolidinedione (TZD) derivative [64]. The yeast MPC1:MPC3 activity was also sensitive to Zaprinast (a phosphodiesterase inhibitor) and to 7ACC2 and Lonidamine (both are monocarboxylate transporter inhibitors), but not to pioglitazone (another TZD) [35].

Gray and colleagues recently introduced a streamlined plate assay to measure mitochondrial pyruvate carrier activity [32]. By studying pyruvate internalization in mouse and rat liver extracts and comparing to the available literature, the authors noted a wide range of MPC-related transport activity values, largely when extracted mitochondria were used. They concluded that a multitude of parameters, such as (i) the source of isolated mitochondria (species and tissue), (ii) the assay temperature, (iii) the choice of a buffering system as the source of protons (to be co-transported with pyruvate), (iv) the most often unknown absolute amount of MPC contained in total mitochondria protein, and even (iv) the basal metabolic rates of the source organism, could all contribute to significant differences in kinetic parameters [32]. 

For instance, transport studies with the reconstituted purified MPC (from human and yeast) were performed at room temperature [31,35], whereas the work with the rat liver mitochondrial extract was conducted at 6 °C [7]. Assaying pyruvate transport at room temperature, Gray and colleagues determined the rapid uptake for mouse liver (saturation around 30 s), which was slower for the rat liver counterpart, with K_m_ in the range of 0.03 mM–0.07 mM, respectively. Maximum velocities were about 1.1 (mouse liver) and 1.4 (rat liver) nmol.min^−1^ per milligram of mitochondrial protein [32]. Therefore, in the absence of a unified protocol to measure pyruvate transport, the published kinetic parameters for MPC, including those for the purified and reconstituted proteins, can at best be compared by their order magnitudes, rather than by their margin of uncertainty.

Altogether, besides the presumed evolutionary relationship between MPC and dimeric SemiSWEET [19], the only experimental evidence that points to a similar dimeric MPC complex, regardless of subunit combination (hetero- or homotypic), is taken from SEC–MALS studies of protein–detergent complexes [35]. To interrogate this further, we used chemical crosslinking in isolated mitochondria, carefully titrated as a function of concentration and incubation time, to prevent non-specific events. We chose to perform the homobifunctional crosslinking of primary amine groups, which are mainly found in the several lysine side chains (seven in total) scattered across the hydrophilic (and potentially disordered) regions of MPC2; these include two lysine residues in the two looping regions that connect the three TMH (Figure 1). In this way, we verified the formation of MPC2 oligomers, starting from dimers up to higher-order molecular species [31]. In DDM micelles, on the other hand, we did not observe MPC2 species higher than dimers, even at the exceedingly high protein and crosslinker doses, reinforcing the evidence of the monomeric state of MPC2 in the PDC. By contrast, when DDM-solubilized MPC2 was reconstituted into asolectin vesicles, the high-order crosslinking pattern was fully recovered [31]. These results suggest that recombinant MPC2 in a mimetic lipid bilayer, which was proven functional for pyruvate transport, shares high-order oligomeric similarities to the native mitochondrial membrane environment. The ladder-like migration pattern of MPC2 in a gel [31] is a known characteristic of the partial crosslinking of actual interacting multimers; given sufficient incubation time and molar excess of the crosslinker, complete conjugation should be achieved, and the correct size of the complex thus revealed. Since we did not seek full conjugation in the previously published experiment [31], the accurate stoichiometry of human MPC2 remains open to investigation.

In this sense, while the successful isolation of pure MPC heterodimers and homodimers in protein–detergent complexes represents an enormous technical advance [35,36], one cannot exclude the possibility that dimerization itself may simply be an artifact of the PDC, possibly due to lack of hydrophobic matching between the transmembrane helices and the detergent tails [65]; i.e., the higher-order complex will self-assemble upon insertion in a lipid bilayer. In the same context, the possible effects of C12E8 on the destabilization of the human MPC1:MPC2 upon extraction from the mitochondrial membrane [31] deserve further investigation. It is well known that the direct association between detergent molecules and membrane proteins may have disrupting effects on the intermolecular contacts that hold oligomeric proteins together while maintaining solubility [66].

Therefore, based on all the data reviewed above, we hypothesize that a conclusive analysis of size and stoichiometry can only be performed when MPC is studied in its native membrane, under physiologically relevant conditions. Gene editing technologies, such as the CRISPR system [67], can allow for the targeted insertion of C-terminal affinity or fluorescent fusion tags in the genome while maintaining endogenous MPC expression levels in a homologous cell system. Styrene maleic acid (SMA) copolymers can then be used to directly extract MPC from isolated mitochondria [68]. The amphipathic nature of SMA favors its insertion into biological membranes, resulting in the extraction of small discs of lipid bilayer, typically containing integral membrane proteins, along with intrinsic and annular lipids, thus allowing for subsequent chromatographic purification in an aqueous environment [69,70], or by SMA-PAGE [71], hence eliminating the need for traditional detergents. The SMA–lipid–protein (SMALP) particles can be directly studied by a structural technique such as cryo-EM (http://www.smalp.net, [68,72]), and the accompanying lipids can be identified by targeted lipidomics [73].

Though this combined methodological approach is still unavailable in general, we have meanwhile extracted human MPC2-EGFP from our yeast mitochondrial system [31] into SMALP particles. The MPC–SMA–lipid particles were purified by immobilized metal affinity chromatography and size exclusion (Figure 2a). The peak fractions containing the highest fluorescence signal (labeled 1 to 3) were pooled and used for grid setup and submitted to qualitative cryo-EM analysis (Figure 2b). Due to a high degree of contamination, as observed by SDS-PAGE, we did not test the fractions 4 to 6 that belonged to the second peak (Figure 2b).

Consistently, the two-dimensional classification of extracted particles produced high-quality averaging, aligned at relatively high resolutions between 6 and 15 Å (Figure 2c). However, the classes were dominated by the EGFP tags, with a noticeable conformational variability, possibly owing to the long linker peptide to MPC2 C-terminal. The structural disorder and the relatively large EGFP tags hampered any definite conclusions on the MPC–SMA–lipid particles, including an interpretable three-dimensional reconstruction (not shown). On the other hand, the presence of a consistent number of apparently four adjoining EGFP particles, per class, strongly reinforced previous evidence that MPC2 can form high-order when in lipid bilayer [31], thus allowing for a schematic prediction of the SMALP (Figure 2d). To a much smaller extent (data not shown), classes with apparently three or five EGFP particles were also obtained. Given the possibility of projection or averaging artifacts and their dominant sizes, the EGFP tags should be removed before future cryo-grid preparations to produce a more conclusive description of the structural organization of the MPC2 homomer. Additionally, even though SMAL formation has been reported to preserve the native oligomeric state of membrane proteins [68], we cannot exclude the possibility that two independent MPC2 dimers, or even four independent MPC2 monomers, are co-situated within the SMALP. Additionally, the protein’s functionality within the SMALPS (for example, by binding to known MPC inhibitors) has not yet been assessed.

Ever since an evolutionary relationship between MPC and bacterial SemiSWEET was first proposed [19], homology modeling has been widely used to elaborate on the structural rationale of MPC hetero- [9,12,21,33,34] and, more recently, homodimers [36]. However, in these structural analyses, MPC1 remains a 3TMH protein, possibly due to biased modeling algorithms. Concerns on this and several other inconsistencies have been raised and suggest that the analogy made between MPC and SemiSWEET [27] may be faulty and simplistic.

α-helical transporters typically have in common a subset of amphiphilic helices that bundle together to stabilize helix–helix interaction and can be either fully transmembrane or only re-entrant [74]. This creates the cavity (or pore) through which the often-polar cognate substrate may pass. Computational approaches that can efficiently detect transmembrane helices with the presence of pore-lining residues are able to predict pore stoichiometry to provide valuable insight when the overall structure of the protein is unknown. 

Currently, the secondary structure analysis of MPC predicts two TMH for MPC1 and three TMH for MPC2 (or MPC3 in yeast), corresponding to the experimental results by Bender and colleagues [25]. This analysis is based on the primary sequence using various software platforms and made possible by the immense knowledge accumulated in membrane proteins’ structural databases over recent years (https://blanco.biomol.uci.edu/mpstruc). Within its current capability, MEMSAT-SVM [41,42], which employs a supervised learning approach onto the outputs of Pore-Walker [75], likewise predicts the 2TMH and 3TMH topologies for human MPC1 and MPC2, respectively. However, only the C-terminal-most transmembrane helix in human MPC2 (TMH 3, Trp94-Phe112, Figure 1) is expected to be involved in pore formation. Additionally, a pore stoichiometry of four identical helices is proposed by MEMSAT-SVM for MPC2, while MPC1 has only two transmembrane helices and none are pore-facing (yellow circles framed in red in Figure 1). This seemingly makes MPC1 incapable of contributing to a functional pore. 

In principle, the previously proposed heterodimeric MPC1:MPC2 complex would contain only five helices in total (outer plus cavity-lining helices), which is unlikely in terms of cavity dimensions and composition. To our knowledge, there are no other transmembrane transporters across the SLC superfamily—monomeric or oligomeric—that have only five TMH in total; the minimum necessary, for the smallest carriers, appears to be six TMH (e.g., bacterial SemiSWEET). 

As with any other computational prediction, this conclusion must be taken with caution, especially in the case of MPC. However, this prediction agrees with our preliminary cryo-EM analysis (Figure 2c), which suggested the inclusion of about four MPC2 monomers per SMAL particle. Each monomer may contribute its C-terminal helix to a parallel pore arranged with rotational symmetry. Moreover, this model does not exclude the direct interaction between MPC1 and MCP2 within the membrane layer. This only suggests that, when heterocomplexes are necessary, MPC1 may act like an accessory or regulatory protein that allows for the complex stabilization, regulation, and improved efficiency, while MPC2 forms the pore. Further research is required to settle this question *in vitro* and *in vivo*.

Remarkably, Oonthonpan and colleagues recently reported two pathological mutations in human MPC1 that cause developmental abnormalities, neurological problems, metabolic deficits, and for one patient, early death [43]. In particular, the authors determined that the Leu79His substitution eliminated pyruvate transport while still allowing complex formation with MPC2. The same point mutation was reported in the inaugural work on MPC by Bricker and colleagues [13], alongside variation Arg97Trp, in a family of patients that suffered a defect in mitochondrial pyruvate oxidation. The authors noted that both residues are conserved across species [13]. However, the experimental observation that MPC1 has only two TMH [25], combined with secondary structure predictions at online servers that imply that the second TMH ends around amino acid Tyr71, suggests that these mutations occur in a more hydrophilic, non-transmembrane segment of MPC1 that is associated with the mitochondrial matrix [25], thus unlikely to participate in pore formation (Figure 1).

The same analogy can be made for the point mutant Asp118Glu in yeast MPC1 that provided resistance to chemical inhibition to UK5099, as also reported by Bricker and co-workers [13], which lies adjacent to the C-terminal end of this protein. Importantly, Asp118 in yeast MPC1 does not have a corresponding residue in human MPC1, which is only 109 residues in total. Therefore, the purportedly disordered regions that flank the transmembrane helices, which are the source of significant sequence variability among species (length and composition-wise), may also play a crucial role in regulating active pyruvate transport by MPC by still elusive mechanisms, thus warranting further investigation. Interestingly, several post-translation modifications in hydrophilic segments of human MPC1 (Figure 1), such as lysine acetylation (at Lys45, Lys46, and Lys72), and phosphorylation (at Tyr12, Ser15, and His101), have been experimentally confirmed in high-throughput proteomic analyses and should also be studied in the context of transport function [44,45,46].

To the best of our knowledge, for human MPC2, neither single-residue pathogenic variants nor post-translational modifications have been reported thus far. On the other hand, binary interactions between MPC2 and ten other proteins have been experimentally detected by prey pooling in two-hybrid approaches, for which a network is available at The Human Reference Protein Interactome Mapping Project (http://www.interactome-atlas.org). No interactors with MPC1, other than MPC2 itself, have been currently identified.

## 4. Conclusions

The functional link between the oligomeric state of MPC and pyruvate transport remains a burning question with, to date, many conflicting research outcomes. These may be explained by the variation in experimental design, which is intrinsic to each step in the study of membrane proteins. From expression and purification to functional reconstitution in lipid vesicles, there are many decision points in this process that may predispose the eventual results. All the reviewed studies above are valid within their limitations and greatly contribute to the knowledge of MPC. Nonetheless, when taken together, they make clear that, at least for the time being, one cannot draw general conclusions about the properties of MPC in eukaryotes by simple analogy to the yeast or the human systems alone. However, an emerging picture is that MPC1, with 2 TMH, may play structural and regulatory roles within the heterocomplex, rather than core substrate transport activity. Although we showed that human MPC2 could have an autonomous transport activity *in vitro* and a yeast cell model [31], the results of Tavoulari and colleagues demonstrate significantly higher efficiency for the heterocomplex [35], that could be, in fact, more closely related to the metabolic demands of the cell [7,66].

Nevertheless, the dimeric assembly observed in detergent:lipid particles [35] may be insufficient in size to allow the formation of a fully functional cavity, as it is proposed to contain only five TMH in total [25], and thus higher-order complexes within the lipid membrane bilayer, such as those for MPC2 alone [31] should be expected; these questions deserve a full investigation. Ultimately, perhaps more important than developing equivalent *in vitro* experimental systems is to study each of these proteins’ structure and function in their respective native mitochondria. Excitingly, many emerging experimental approaches (e.g., CRISPR + SMALP + cryo-EM) will make this possible in the short term.

## 5. Materials and Methods

### Formation of SMALP and Cryo-EM Analysis

Membrane pellets containing human MPC2–EGFP were resuspended to 50 mg mL^−1^ in buffer A (50 mM Tris-HCl pH 8, 0.25 M NaCl, 10% glycerol) directly added by the SMA solution XIRAN^®^ SL 30,010 P20 (Polyscope Polymers B. V.) to a final concentration of 2.5%. After incubation for 3 h, continuously stirring at room temperature, the solution was centrifuged at 185,500 × *g* for 45 min to remove un-solubilized particles. The supernatant was incubated overnight at 4 °C with Cobalt or Nickel beads (5 mL resin per gram of membrane pellet) previously equilibrated with buffer B (50 mM Tris-HCl pH 8, 0.15 M NaCl, 3% glycerol). Affinity chromatography was completed by gravity washing resin with 20 column volumes (CV) of buffer B, followed by 5 CV of buffer B supplemented by 20 mM imidazole, and then eluted with 2 CV of Buffer B containing 200 mM Imidazole. The dropwise collected fractions were pooled and concentrated for injection into a Superdex 200 10/300 column previously equilibrated with 50 mM Tris pH 8.0, 150 mM NaCl. For cryo-EM analysis, peak fractions 1–3 (Figure 2b) were pooled and diluted to a concentration of 0.1 mg mL^−1^. A 3 µL drop was dispensed onto the previously glow-discharged grid, blotted with a FEI Vitrobot Mark IV, and flash-frozen in liquid ethane. Cryo-EM grids were loaded into a FEI Titan Krios G3 electron microscope, equipped with a Volta phase plate, operated at 300 kV. Multi-frame exposures were recorded using a Falcon 3 camera. Spherical aberration coefficient (cs) was 2.7 mm, with a nominal magnification of 75,000 times, and a calibrated physical pixel size of 1.08 Å. The values for the second condenser aperture and the illumination area were 50 and 1.8 µm, respectively. The Defocus range was established at −0.7 µm, and the dose rate on specimen 0.7 e-.pix^−1^.s^−1^, fractionated for 75 frames, for a total of 60 s. Movie fractions were aligned by MotionCor2 [76]. Contrast transfer functions were estimated from aligned dose-weighted micrographs using cryoSPARC [77], which was also used for blob-based particle picking, extraction, and 2D classification. 

## Figures and Tables

**Figure 1 biology-09-00407-f001:**
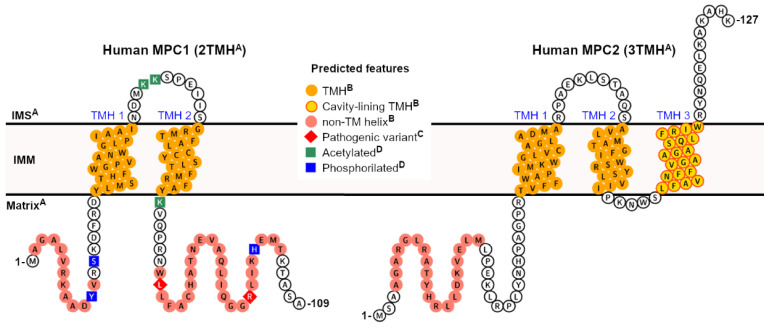
Predicted features of human mitochondrial pyruvate carrier (MPC). A collection of elements, from either direct or correlational pieces of evidence and *in silico* predictions, can be projected for the two human MPC subunits. **^A^** Correlational evidence: the topology and the orientation of each subunit within the IMM bilayer can be inferred from the experimental study on yeast MPC [25], based on significant primary sequence identity between homologs, particularly within the transmembrane helices; **^B^** computational prediction: the presence and positional boundaries of the TMH (orange circles) were predicted by the Constrained Consensus Topology server (CCTOP, [39]), and the non-TM helices (light red circles) by PSI-PRED workbench [40]. Of all five collective transmembrane helices in MPC1 and MPC2, only TMH 3 in MPC2 was predicted to contribute to cavity formation (yellow circles framed in red), according to MEMSAT-SVM [41,42]; **^C^** Experimental evidence: pathogenic variants (red diamonds) that severely compromise pyruvate transport activity have been described in patients [43], while still allowing for heterocomplex formation. For instance, mutations such as Leu79His and Arg97, have been described exclusively in MPC1, although at non-transmembrane regions [43]; **^D^** Experimental evidence: post-translational modifications, such as lysine acetylation (green squares), and phosphorylation (blue squares), have been identified exclusively in human MPC1, in a high-throughput proteomics study [44,45,46]. No signal peptides could be identified by primary sequence analysis.

**Figure 2 biology-09-00407-f002:**
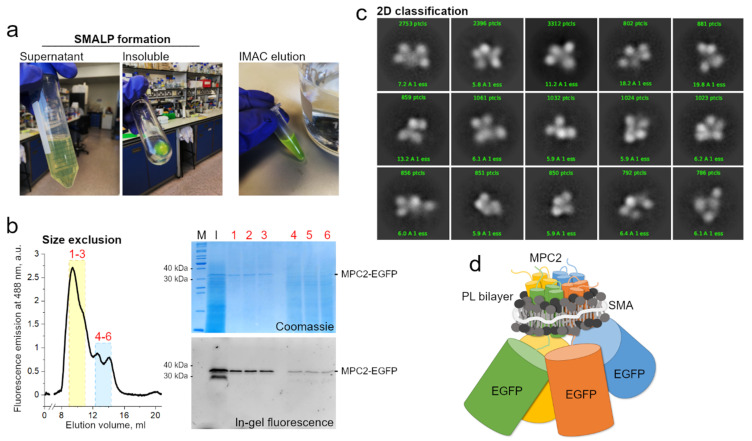
MPC2–EGFP reconstitution into styrene maleic acid (SMA) lipid particles and cryogenic electron microscopy (cryo-EM) analysis. (**a**) SMA encapsulation of MPC2–EGFP showing green fluorescence during initial purification steps. (**b**) Size exclusion chromatography of MPC2–EGFP encapsulated in SMA. Peaks correspond to fluorescence emission at 488 nm. According to column calibration, the dominant peak was observed at an elution volume that corresponded to a molecular weight of around 300–400 KDa. Corresponding eluted peaks showing bands of 30 and 40 kDa for the MPC2–EGFP fusion in both Coomassie and fluorescence SDS-PAGE. (**c**) A two-dimensional classification was applied to the extracted MPC2–EGFP–SMA–lipid–protein (SMALP) particles. (**d**) Schematic representation of the predicted SMA-encapsulated human MPC2–GFP complex.

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
