# Peer review of "Insights on the Quest for the Structure–Function Relationship of the Mitochondrial Pyruvate Carrier"

_biology, 2020, doi:10.3390/biology9110407_

Round 1

Reviewer 1 Report

In this review, Quesñay and co-workers discuss findings from a limited number of recent studies on the mechanism, complex composition and oligomeric state of the mitochondrial pyruvate carrier. The authors focus primarily on three biochemical studies, including one of their own, which use purified MPC proteins to study complex composition and oligomeric state. Additionally, they provide some preliminary experimental data, aiming to support their previous hypothesis that the mitochondrial pyruvate carrier is a higher oligomer of MPC proteins. Interestingly, they also convey a hypothesis that within the MPC complex, MPC2 is responsible for pore formation whereas MPC1 may act as an accessory or regulatory protein but does not participate in substrate translocation.

The key problem is that this paper is neither a review nor a scientific paper. The review part is incomplete and selective, whereas the scientific paper parts represents a lot of preliminary data and claims that are not supported by the data, and thus are nothing but speculation. 

The authors need to revise the manuscript to adopt a more balanced approach when interpreting the available literature and discussing/ supporting their views and experimental work. More importantly, they need to provide additional, conclusive experimental evidence to support any new, unproven hypotheses.

Major comments:
1. MPC functional properties. As the authors state, extensive studies had been performed in the past to elucidate the functional properties of the mitochondrial pyruvate carrier, using systems such as isolated mitochondria from mammalian species where the protein is in its natural environment. These studies provide a wealth of information on substrate binding and transport, as well as inhibition of MPC by small molecules. Throughout the paper, the authors highlight the importance of studying MPC in the context of the membrane rather than in detergent/lipid complexes. However, they have excluded these studies from their discussion and from the evaluation of the recent findings. They need to describe how recent publications on the MPC proteins, including their own, relate to the previously published MPC functional characteristics. How have purified proteins behaved compared to what is expected from the studies in isolated mitochondria? Does transport have similar rates in comparable temperatures? Previous studies suggest very rapid, in the time frame of seconds, pyruvate transport in isolated mitochondria at room temperature but some reconstituted systems require several minutes to detect uptake. How do kinetic characteristics compare? Does transport get inhibited by previously established MPC inhibitors and if so, is the potency or affinity comparable to what is known?

The authors have previously introduced in the literature the notion that MPC2 alone is a functional mitochondrial pyruvate carrier. How do MPC2 functional properties compare to those of the MPC1/MPC2 hetero-complex in their systems or compared to other purified proteins in detergent or isolated mitochondria?

2. Composition of the functional complex. There is a consensus in the literature, supported by several lines of experimental evidence, provided by different research groups that formation of the MPC1/MPC2 hetero-complex supports mitochondrial pyruvate transport. The authors have previously proposed that additionally to MPC1/MPC2 hetero-complex, MPC2 alone is also active. Here, they progress their hypothesis to propose that within the hetero-complex transport is mediated via MPC2, whereas MPC1 holds an accessory or regulatory role. No experimental data is provided to that end. The argument is based on in silico predictions suggesting that only the C-terminal-most TMH of MPC2 can participate in pore formation but not MPC1. On the contrary, several studies suggest that both proteins are necessary for pyruvate transport but some of them are not even referenced in this review. For example, human pathological mutations have been found in the MPC1 protein. One of them in particular, L79H, eliminates pyruvate transport while allowing complex formation, suggesting MPC1 is also critical for substrate translocation.

It is important that the authors provide conclusive experimental evidence if they want to make this new claim. At a minimum, they should provide a comparative transport assessment between the hetero-complex MPC1/MPC2 and MPC2 homomers. For example, this could be done in their in vivo yeast system and yeast isolated mitochondria where they previously showed activity for MPC2 but not in comparison to MPC1/MPC2.

Lastly, the authors do not discuss possible downsides of their own previous work on this topic. For example, their strategy to purify the human MPC hetero-complex involved GFP tagging that dominates the MPC2 protein. They have not discussed the possibility that this tag disrupts the hetero-complex interface, not allowing co-purification of the two proteins, which has been successful by two independent groups for both yeast and human MPC.

3. Oligomeric state. The authors claim that in the membrane environment MPC2 forms higher oligomers. In their previous work higher oligomers had been detected upon application of chemical cross-linking. First, applying chemical cross-linkers is not necessarily physiologically relevant. Second, even in the presence of high concentrations of cross-linkers their samples are dominated by monomers. Here, they purify EGFP-tagged MPC2 to find that the protein is again a monomer in DDM or UDM. Upon isolation of the protein in SMALPs followed by Cryo-EM, they find a range of oligomeric states, they do not show reproducible structural order (see points further below). Moreover, functionality of the protein within the SMALPS (i.e. binding of known MPC inhibitors) has not been assessed and needs to be included to exclude the possibility that they are just aggregates of different sizes, monomer, dimer, trimer and terramer. Furtermore, the authors do not explain how they exclude the possibility that what they see in SMALPS is not the functional unit. How do they exclude the possibility that two dimers or four monomers are situated within the SMALP?

The authors should not draw conclusions on the oligomeric state of the hetero-complex, which they have not studied. Although they do not dispute the validity of the SEC-MALLS results for the yeast hetero-complex, they claim that hetero-dimer formation is an artefact of the preferred detergent/lipid mix. The authors do not have evidence to support this criticism and this hetero-dimer has been reconstituted and was shown to transport pyruvate with properties of the native transporter. This and other unproven claims mentioned above have to be removed from the manuscript.

4. Structural analysis. Size exclusion chromatography shows that the MPC2-EGFP encapsulated in SMA is not monodisperse, eluting in a broad range of sizes, indicative of aggregation (Figure 2). In agreement, the 2D classification report, which might be of the fraction 1,2 and/or 3, shows different assemblies that are not structurally related to each other, and thus could represent random aggregates. The 2D classes are low-resolution, meaning a crude assembly of largely disordered structures. Moreover, there are no 3D reconstructions to show reproducible structural order between 2D classes. These analyses do not prove at all that MPC2 forms reproducible tetrameric arrangements, as they might be from fractions of the size exclusion chromatography enriched for these higher molecular weight aggregates. What about fraction 4-6? They might well contain monomers or dimers. Membrane proteins, which are ill-treated, are often found in different states of aggregation, which have nothing to do with the functional state. See also sizeable fluorescent pellet in 2a, indicating a large aggregate.

The results shown in Figure 1 show lack of progress in structural analysis. Figure 1b clearly shows detergent crystals, which form low-resolution diffracting crystals with a hexagonal packing arrangement, because of close-packing of spherical micelles without internal order. The crystals in Figure 1C contain protein (fluorescence), but have no order at all. Thus, these results provide no evidence for the claims made there is some kind of defined structure of the isolated protein. This is all very preliminary and inconclusive. Why show this? Nothing can be concluded from them and thus they should be removed, otherwise it is misleading.

This review needs a better description of past published work and removal of the unsubstantiated claims using new unvalidated data.

Reviewer 2 Report

Dear authors,

I read your review on the structure-function studies of the mitochondrial pyruvat carrier with great interest.

Its content is comprehensive and of interest to a wider readership.

Here are my comments/suggestions:

line 168: a summary figure displaying the different structure-function models of the pyruvat carrier might here - at the end of section 2 - be very helpful for the reader

Figure 1 could instead be deleted as it does not add more information above to what is stated in the text anyway about these crystallisation attempts.

line 242: please introduce SEC-MALS (combination of Size Exclusion Chromatography with Multi-Angle Light Scattering analysis) to the reader

line 288 ref 31: was the reconstituted MPC2 carrier active or was activity not tested?

line 302: please introduce rational behind the styrene maleic acid SMA copolymer technology to the reader

line 309: the context of the first part of the sentence "Though this combined methodological approach is still unavailable..." is not clear (ie not available in your group or generally not available?)

Fig.2 please include the type of SEC column (2b) in the figure legend and state to which MW the elution volumes corrsponde

line 351 please introduce MEMSAT-SVM to the reader

lines 365-376: I think the summary section is too general. I would expect here a set of conclusions/suggestions/models drawn from the data so nicely summarised in the remaining sections. One possible theme that runs through the MPC1 story is that MPC1 with its - possibly - 2 TMHs might have regulatory/structural functions rather than core transport activities.

Round 2

Reviewer 1 Report

The authors have made significant improvements in their manuscript. A small number of issues still remains.

Line 336: “Noteworthy, MPC1 was recently proposed to be the bona fide target of cinnamates [13]”.

More recently, Yamashita et al., 2019 showed that Cys54 in MPC2 interacts with UK5099 and derivatives in a covalent reversible manner. The authors need to include this important information in this context.

Lines 349-351: It is more sensible to state the transport results (time course and kinetic measurements) from the study by Grey et al., and compare with pyruvate transport in MPC proteoliposomes. In the Grey et al., study, pyruvate transport has been performed at room temperature. This provides a good reference point for comparison with pyruvate transport in MPC proteoliposomes, which has also been tested at room temperature, both for yeast and human proteins.

356-358: “…the only experimental evidence that points to a similar dimeric MPC complex, regardless of subunit combination, are taken from protein-detergent complexes and electrophoretic migration [14,19,20]”. This statement is not accurate because it excludes a SEC-MALLS study on the oligomeric state of the yeast MPC. The authors need to add this here.

Lines 364-367: “In DDM micelles, only dimers could be observed, even at the exceedingly high protein and crosslinker doses.”  First, this is not accurate as more monomers than dimers are shown in DDM-purified MPC2 in the presence of cross-linker in the referenced study. In any case, what is the authors’ conclusion; does MPC2 exist as a monomer or a dimer in DDM? It seems that MPC2 is purified as a monomer in DDM, as assessed by SEC-MALS, but when it is cross-linked in DDM it forms dimers. Isn’t that an indication that chemical cross-linking can alter a pre-existing oligomeric state? It would be appropriate that the authors explain which result they trust and why.
